# A Retrospective Comparison of DLI and gDLI for Post-Transplant Treatment

**DOI:** 10.3390/jcm9072204

**Published:** 2020-07-12

**Authors:** Sylvain Lamure, Franciane Paul, Anne-Laure Gagez, Jérémy Delage, Laure Vincent, Nathalie Fegueux, Anne Sirvent, Eve Gehlkopf, Jean Luc Veyrune, Lu Zhao Yang, Tarik Kanouni, Valère Cacheux, Jérôme Moreaux, Beatrice Bonafoux, Guillaume Cartron, John De Vos, Patrice Ceballos

**Affiliations:** 1Department of Clinical Haematology, CHU Montpellier, 34295 Montpellier, France; s-lamure@chu-montpellier.fr (S.L.); f-paul@chu-montpellier.fr (F.P.); algagez@santelys.fr (A.-L.G.); j-delage@chu-montpellier.fr (J.D.); l-vincent@chu-montpellier.fr (L.V.); n-fegueux@chu-montpellier.fr (N.F.); a-sirvent@chu-montpellier.fr (A.S.); e-gehlkopf@chu-montpellier.fr (E.G.); t-kanouni@chu-montpellier.fr (T.K.); g-cartron@chu-montpellier.fr (G.C.); 2Federation of Haematology, University of Montpellier, 34295 Montpellier, France; 3UMR-CNRS 5535, Institut de Génétique Moléculaire de Montpellier, 34090 Montpellier, France; 4Direction de la Stratégie, des Affaires Médicales et de l’Innovation, Santélys Association, 59120 Loos, France; 5Clinique du Parc, 34170 Castelnau le Lez, France; 6Department of Cell and Tissue Engineering, University of Montpellier, CHU Montpellier, 34295 Montpellier, France; jl-veyrune@chu-montpellier.fr (J.L.V.); zy-lu@chu-montpellier.fr (L.Z.Y.); 7Department of Biological Haematology, CHU Montpellier, 34295 Montpellier, France; v-cacheux@chu-montpellier.fr (V.C.); jerome.moreaux@igh.cnrs.fr (J.M.); 8IGH, CNRS, University of Montpellier, 34094 Montpellier, France; 9Institut Universitaire de France, 75005 Paris, France; 10Department of Immunology, CHU Montpellier, 34295 Montpellier, France; b-bonafoux@chu-montpellier.fr

**Keywords:** donor lymphocyte infusion, allogeneic stem cell transplantation, post-transplant treatment

## Abstract

Donor lymphocyte infusion (DLI) is used to prevent or treat haematological malignancies relapse after allogeneic stem cell transplantation (allo-SCT). Recombinant human granulocyte colony-stimulated factor primed DLI (gDLI) is derived from frozen aliquots of the peripheral blood stem cell collection. We compared the efficacy and safety of gDLI and classical DLI after allo-SCT. We excluded haploidentical allo-SCT. Initial diseases were acute myeloblastic leukaemia (*n* = 45), myeloma (*n* = 38), acute lymphoblastic leukaemia (*n* = 20), non-Hodgkin lymphoma (*n* = 10), myelodysplasia (*n* = 8), Hodgkin lymphoma (*n* = 8), chronic lymphocytic leukaemia (*n* = 7), chronic myeloid leukaemia (*n* = 2) and osteomyelofibrosis (*n* = 1). Indications for DLI were relapse (*n* = 96) or pre-emptive treatment (*n* = 43). Sixty-eight patients had classical DLI and 71 had gDLI. The response rate was 38.2%, the 5-year progression-free survival (PFS) rate was 38% (29–48) and the 5-year overall survival (OS) rate was 37% (29–47). Graft versus host disease rate was 46.7% and 10.1% of patients died from toxicity. There were no differences between classical DLI and gDLI in terms of response (*p* = 0.28), 5-year PFS (*p* = 0.90), 5-year OS (*p*. 0.50), GvHD (*p* = 0.86), treated GvHD (*p* = 0.81) and cause of mortality (*p*. 0.14). In conclusion, this study points out no major effectiveness or toxicity of gDLI compared to classical DLI.

## 1. Introduction

Donor lymphocyte infusion (DLI) can be used to prevent or cure haematological malignancies relapse after allogeneic stem cell transplantation (allo-SCT) using the antitumoral effect of donor T cells. Efficiency was shown first in chronic myeloid leukaemia (CML) and indolent lymphoma [1]. Patients with acute myeloid leukaemia (AML), myelodysplasia or acute lymphoid leukaemia (ALL) have a poorer response to DLI and patients with multiple myeloma could benefit from this procedure [2]. DLI therapeutic action is due to the antitumoral activity of T lymphocytes, involving T cell exhaustion reversal and T cells infiltrating tumour activity [3]. The efficacy of DLI is correlated with a low tumour burden [2]. Mixed chimerism and detectable minimal residual disease reflect these low tumour burden situations, preceding clinical relapse and raising the interest of pre-emptive use of DLI in these situations [4]. In return, DLI induces acute and chronic graft versus host disease (GvHD) in 30% and 44%, respectively [5]. This procedure is effective alone or associated with antitumoral chemotherapy or targeted therapy [6,7,8,9].

Recombinant human granulocyte colony-stimulated factor (G-CSF) primed DLI (gDLI) derives from the frozen surplus of the initial peripheral blood stem cells (PBSC) bag collection. PBSC collection often exceeds the required amount for transplantation (5 to 10 × 10^6^/kg) and excess can be cryopreserved. PBSC grafts contain T lymphocytes that can endure cryopreservation. Based on the antitumoral effect of T lymphocytes infused with PBSC, efficacy and safety of gDLI have been described in different clinical situations [10,11,12,13]. However, there are multiple biological effects of G-CSF on peripheral T cells that could potentially alter their biological functions: polarization T cells from Th1 to Th2, promotion of regulatory T cells and induction of tolerogenic dendritic cell differentiation [14,15]. 

In this monocentric retrospective study, we aim to compare the efficacy and toxicity of gDLI and classical DLI in a continuous cohort of 139 patients treated for haematological malignancies.

## 2. Methods

### 2.1. Patients and Procedure

We performed a systematic, retrospective review of the medical charts of all patients who underwent HLA allo-SCT and DLI for haematological malignancies in Montpellier University Hospital between January 1998 and December 2018, in curative or pre-emptive situations. Each patient provided written informed consent before transplantation and before DLI. Classical criteria were used for response assessment, conditioning definition and GvHD grading [16]. Mixed chimerism was defined as <95% of donor T cells and nucleated cells in peripheral blood three months after transplant. 

Patients treated successively with gDLI and then, after depletion of gDLI bags, with fresh classical DLI were excluded from this study. We excluded patients who received DLI after haplo-identical allo-SCT as well as patients who received prophylactic DLI due to their small number.

Clinical relapse was demonstrated by clinical evaluation, medical imaging, blood or marrow count. Pre-emptive situations were defined either by a decrease (>5%) in chimerism, stable mixed chimerism or detectable minimal residual disease (MRD) by a polymerase chain reaction (PCR) or flow cytometry (according to diseases). Chimerism was determined using the multiplex amplification of short tandem repeat markers and fluorescence detection [17]. 

The DLI groups were established as follows. Classical DLI is defined by the infusion of fresh cells issued from new donor leukapheresis for the first DLI and cryopreserved cells for the following DLI and, in some cases, another collection was performed after all cryopreserved bags were used. gDLI is defined by the infusion of cryopreserved aliquots derived from the PBSC bag used for transplant, for all DLI.

DLI was prescribed at least two months after transplant and after immunosuppressive treatment interruption if the patient showed no active GvHD or infection. Antitumoral treatment may be associated with DLI. Post-DLI GvHD was defined as acute GvHD or overlap syndrome of chronic GvHD. The infused cell amount depended on indication, according to recommendations [18]. Clinical and biological evaluation was carried out every 14 days after injection. The DLI program was interrupted after GvHD or a response. A response was defined according to international criteria. For pre-emptive indications, undetectable MRD and/or total donor chimerism (>95%) were considered as a response. Response was assessed one to three months after infusion and every three months. When new clinical or molecular relapse occurred more than one year after the last DLI injection, a second program could be performed, adapted to the response to the first DLI program. When a second allogeneic stem-cell transplant was performed, including a second DLI program, two different observations were collected and appeared as two patients in the study. Progression-free survival (PFS) and overall survival (OS) were defined as the time between the first DLI and progression or death due to any cause. Toxicity-related mortality combined death due to GvHD or infection following DLI, as those situations frequently overlaps.

### 2.2. Statistical Analysis

Distributions of data were tested with the Shapiro–Wilk test. Chi-squared or Fisher’s exact tests were used to compare categorical data. For numerical data, medians were compared using the Student T test or the Mann–Whitney test. The association between covariates and the patient’s response was assessed using multivariate logistic regression analysis. The receiver operating characteristics (ROC) curve was used to determine the threshold capable of predicting complete response with CD34+ and CD3+ cells numeration associated with the best sensitivity and specificity according to the Youden index. PFS and OS were estimated using the Kaplan–Meier method and comparisons were made using the log-rank test. Hazard ratios and their 95% confidence intervals in univariate and multivariate analyses were calculated using the Cox regression. All statistical analyses were performed at the conventional two-tailed α level of 0.05 using R software version 3.0.2.10.

## 3. Results

### 3.1. Patient and DLI Characteristics

Between 1998 and 2018, 1123 patients received an allogeneic transplant in our centre. A total of 159 patients (14.2%) received a DLI program. Twenty patients were excluded (1 had haploidentical transplant, 5 had prophylactic DLI and 14 received successively gDLI and fresh classical DLI). Three patients had two allogeneic transplants. We therefore considered two DLI programs independently for these three patients. One-hundred and thirty-nine patients were analysed; 68 received classical DLI, while 71 received gDLI, (Figure 1). The median age at transplant was 52 years old (41–61) and 54.7% of patients were men. Indications for transplantation were AML (*n* = 45), myeloma (*n* = 38), ALL (*n* = 20), non-Hodgkin lymphoma (*n* = 10), myelodysplasia (*n* = 8), Hodgkin lymphoma (*n* = 8), Chronic lymphocytic leukaemia (CLL) (*n* = 7), CML (*n* = 2) and osteomyelofibrosis (*n* = 1). The disease status at transplantation was complete remission (CR) for 54.7% of patients, partial remission (PR) for 27.3% and stable or progressive disease for 18.0%. The conditioning regimen was myeloablative (MAC) for 23.2% of patients, reduced intensity (RIC) for 66.7% and sequential for 10.1%. Transplant was issued from a family-related donor for 48.9% of patients, from a matched unrelated donor for 51.1%, or from an unrelated HLA mismatch for 12.2%. The graft origin was peripheral blood stem cells for 82.7% of patients. The median infused CD34+ cell count was 5 × 10^6^/kg (IQR 4.4–5.5) and the median infused CD3+ cell count was 16.9 × 10^7^/kg with a higher heterogeneity (IQR 9.6–23). The indications for DLI were relapse (69.1%) or pre-emptive situation (mixed chimerism, 15.8%; detectable MRD, 15.1%). The number of DLIs was more than three for 10.8% of patients. The number of patients treated with DLIs increased over time: 18 between 1998 and 2004, 59 between 2005 and 2011 and 62 between 2012 and 2018. DLI was associated with another curative treatment in 34.6% of patients: tyrosine kinase inhibitor for CML or ALL (*n* = 10), hypomethylating agents for myelodysplasia or AML (*n* = 24), immunomodulatory drug for myeloma (*n* = 15), anti-CD20 monoclonal antibody for non-Hodgkin lymphoma (*n* = 1), anti-CD33 or anti-CD30 monoclonal antibody-drug conjugates for AML and Hodgkin lymphoma, respectively (*n* = 1 each), and JAK2 inhibitor for osteomyelofibrosis (*n* = 1) or chemotherapy (*n* = 6). Acute GvHD occurred in 79.9% of patients: 72.7% had grade I, 19.1% grade II and 8.2% grade III; and chronic GvHD occurred in 15.9% of patients. Three-month mixed chimerism was observed for 58.7% of patients. We observed no autologous recovery. Median follow-up was 21.7 months, and median follow-up for survival cases was 55.6 months.

Median age was 49 (24–58) in the classical DLI group vs. 55 (47–63) in the gDLI group, *p* < 0.01. Classical DLIs were the only cell product used before 2007, hence there are more patients treated for CLL and ALL (*p* = 0.05) and with myeloablative conditioning (*p* = 0.02) in the classical DLI group. Classical DLI patients were also more likely to receive a related transplant (*p* < 0.001). In accordance with the procedure, all gDLI patients had peripheral blood stem cells for transplant (*p* < 0.0001). There were more patients with previous chronic GvHD (cGvHD) in the classical DLI group (22.1% vs. 9.9%, *p* = 0.04). There was no difference in HLA compatibility, hematopoietic stem cell (CD34+) amount, T lymphocyte (CD3+) amount in transplant, acute GvHD, chimerism evaluation, indication for DLI or three or more DLIs (Table 1).

Two-hundred and sixty-eight DLI units were administered: 72 patients received a single dose, 33 received two doses, 19 received three doses, 8 received four doses, 2 patients received five, 4 patients received six doses and 1 patient received seven doses. The median time for the first DLI was 8.5 months, 5.9 months between DLI 1 and 2 and 12.7 months between DLI 2 and 3. The median dose of CD3+ T lymphocyte was 1 × 10^7^/kg for the first DLI (DLI1), 1.4 × 10^7^/kg for DLI2 and 2.2 × 10^7^/kg for DLI3 (Table 2). The median time for the first DLI was 12 days shorter in the gDLI group compared to classical DLI.

### 3.2. Response

Out of all of the patients treated with DLI, complete response was observed in 38.2%. The response rate for gDLI was 43.3% and 32.8% for classical DLI, *p* = 0.28. The response rate for AML was 26.2%, 37.1% for myeloma, 26.3% for ALL, 75% for myelodysplasia, 80% for non-Hodgkin lymphoma, 42.9% for Hodgkin lymphoma, 42.9% for CLL, 50% for CML and no response for osteomyelofibrosis (*p* = 0.02). The response rate for relapse treatment was 26.1%, 42.9% for detectable MRD treatment, and 81.8% for mixed chimerism treatment (*p* < 0.001) (Table 3).

In univariate analysis, Non-Hodgkin Lymphomas (NHL) and myelodysplastic syndromes had a better response rate: OR = 11.28 (2.07–61.44, *p* < 0.01) and OR = 8.46 (1.48–48.26, *p* = 0.02), respectively, compared to AML. Relapse situation was associated with a poor outcome compared to pre-emptive situations: OR = 0.21 (0.09–0.45, *p* < 0.0001). Severe post-DLI GvHD (requiring treatment) was associated with response: OR = 9.08 (3.28–25.14, *p* < 0.0001). These differences were confirmed in the multivariate model. There was no difference between classical DLI and gDLI in terms of response (Table 4).

### 3.3. Safety

DLI induced GvHD (acute GvHD or overlap syndrome) in 46.7% of patients and 23.4% experienced severe GvHD requiring treatment. The GvHD rate was 47.9% in the gDLI group and 45.5% in the classical DLI group (*p* = 0.86). In the gDLI group, the rate of GvHD requiring treatment was 25.4% vs. 21.2% in the classical DLI group (*p* = 0.81). Toxicity-related mortality (TRM), combining GvHD and infections post-DLI, occurred in 10.1% of the patients. Univariate and multivariate analyses show the association between occurrence of treatment requiring GvHD and TRM, HR = 9.6 (2.1–43.7, *p* < 0.01) (Appendix A).

### 3.4. Survival

Considering all patients, the median PFS was 9 months (3–35), 5-year PFS was 38% (29–47), the median OS was 22 months (6–50) and 5-year OS was 37% (29–47). Eighty-two deaths were recorded: 47.1% due to progression of the malignancy, 10.1% due to toxicity of the DLI (GvHD and infection) and three deaths from other causes. There was no difference in these outcomes between the classical DLI and gDLI groups (Table 3). Univariate analysis shows association of ALL, CLL, NHL and MM with a better OS, relapse indication for DLI was associated with a worse OS, HR = 5.06 (2.7–9.6, *p* < 0.01) and post DLI GvHD not requiring treatment was associated with a better OS, HR = 0.5 (0.29–0.87, *p* = 0.01). This last point was not confirmed in the multivariate analysis (Appendix A).

The median PFS was 4 months for AML, 10 months for multiple myeloma, 9 months for ALL, 23 months for MDS, 24 months for CLL, 17 months for Hodgkin lymphoma, 39 months for non-Hodgkin lymphoma, 16 months for CML and 2 months for OMF. The median OS was 6 months for AML, 34 months for multiple myeloma, 21 months for ALL, 27 months for MDS, 62 months for CLL, 17 months for Hodgkin lymphoma, 48 months for non-Hodgkin lymphoma, 17 months for CML and 3 months for OMF. The median PFS and OS for DLI curative indications were 5 months (2–22) and 12 months (4–37), respectively. The median PFS and OS for DLI pre-emptive indications were 30 months (13–67) and 40 months (22–68). (Figure 2 and Figure 3). Univariate analysis shows a better PFS among NHL, HR = 0.23 (0.07–0.76, *p* = 0.02), relapse indication for DLI was associated with a worse PFS, HR = 3.31 (1.87–5.86, *p* < 0.01); post DLI GvHD not requiring treatment and treatment-requiring GvHD were associated with a better PFS, HR = 0.56 (0.33–0.94, *p* = 0.03) and HR = 0.19 (0.08–0.45, *p* < 0.01). All those associations remained significant in the multivariate analysis (Appendix A).

### 3.5. Sub-Group Analysis

We performed sub-group analysis in four different populations. Relapse and pre-emptive indications in one hand, and myeloma and AML in the other hand. Those groups were important enough to reach significance if any. In all four sub-groups, there was no different between gDLI and classical DLI, in terms of complete response rate, post-DLI GVHD and treatment requiring GVHD rate, PFS and OS and cause for mortality (Appendix A).

## 4. Discussion

This large monocentric retrospective continuous cohort study comparing classical DLI and gDLI for haematological malignancies found no noteworthy difference between the two procedures in terms of response, toxicity and outcome.

This study spans more than 20 years of transplant practice, in which DLI indications have increased and the following situation for DLIs has expanded: pre-emptive situations for mixed chimerism and then for detectable MRD [2]. At the same time, haematological stem cell transplants have evolved [19], with the increasing frequency of unrelated transplants, the diversification of HPSC cell sources and reduced-intensity conditioning. The first gDLI was performed in our centre in 2007, and we gradually replaced classical DLI with gDLI when the latter became available. Our study suffers from some bias relating to these changes in practice. The interpretation of this global comparison of gDLI and DLI is also limited by the variability of malignancies. However, subgroup analysis showed no differences in DLI and gDLI efficiency and toxicity for the different diseases. Available data on gDLI bag composition were limited to CD3 and CD34 positive cells. However, in a future study, differential analysis of lymphocytes sub-population composition, as well as NK cell rates of classical DLI and gDLI would be interesting. The availability of gDLI bypasses new donor recruitment, qualification and cell collection. This is a major advantage for organisational reasons and for saving time and costs, especially for DLIs from donors from distant countries. In the gDLI group, the median time to the first injection was 12 days shorter compared to the classical DLI group.

In AML and myelodysplasia, post allo-SCT relapses represent the major pitfall and the prognosis is poor: the response rate to DLI is around 35% and 2 years of OS between 12% and 46% [2]. Our data show outcomes comparable to those reported by others: 26.2% in the AML group, 75% in the myelodysplasia group which is small (eight patients). In ALL, post-DLI 5-year OS is typically 5% to 12%, with a response rate of 25% [20,21]. The data reported here are similar for the response (26.3%) with a better OS, and this may be explained by our frequent pre-emptive strategy (molecular tools for MRD monitoring) as well as the concomitant use of tyrosine kinase inhibitors. In indolent lymphoid malignancies, the DLI response rate varies from 76% to 92% in curative or pre-emptive situations [22]. In Hodgkin lymphoma, pre-emptive DLI for mixed chimerism induces a long-term response [23] and the therapeutic DLI response rate is between 40% and 79%. Altogether, 3-year OS after DLI is between 44% and 59% in lymphoid malignancies [2]. Our data are limited for those malignancies but are consistent with these figures. DLI associated with targeted therapy for myeloma confers a 58–75% response, which is superior to our data (31.1%). However, the reported 3-year OS is 43–73%, which is consistent with our data [6]. Pre-emptive DLI used for mixed chimerism shows a comparable response rate in our experience and in the literature. Indeed, a response rate of 68% and a 2-year OS rate of 80% for transplanted AML [24] and 3-year OS of 70% for ALL [25] in transplanted lymphoid malignancies have been reported. The response rate after DLI for mixed chimerism is 75–95% and 3-year OS is 44 months [23,26]. Overall, the response to DLI and outcome in our study seem to be comparable with the literature, whatever the DLI indication. The main toxicities are acute GvHD and overlap with chronic GvHD. Comparison with the literature for this issue is difficult due to the combined clinical criteria we chose. Acute GvHD after DLI is reported in 4 to 61% of patients and chronic in 20 to 59% [2]. Reported grade 2–4 acute GvHD was at 26% in a recent study, which is in accordance with our rate of GvHD requiring treatment (23.4%) [5]. Cytopenia has been described as one of the major toxicities following DLI. However, this was not a concern in our experience. We observed few cases with mild and transient cytopenia.

In vitro studies showed less antitumoral activity for T cells harvested after G-CSF stimulation [15], in contrast to the literature reports that have tended to document better outcomes with gDLI [27]. Abbi’s study compared the outcome of classical DLI and gDLI in 67 patients (including 70% of myeloid malignancies), 15 of whom with gDLI and 52 with classical DLI. The outcome was globally poor, with a median OS of 6 months, and 62.7% had GvHD and no difference was shown between classical and gDLI [28]. Hossain reported a series of 63 patients (including 36 with myeloid malignancies, 13 with ALL), of which 40 received gDLI and 13 received classical DLI. No difference between classical DLI and gDLI was shown in the entire cohort (median OS 3.8 versus 4.6 months) but the AML/MDS subgroup had a trend for better outcomes with gDLI (4.1 versus 2.7 months, *p* = 0.16); the rate of grade 3–4 GvHD was high (47.6%) [29]. Hasskarl reported 121 cases of gDLI, 52.1% of myeloid malignancies, 14% of ALL, 10.7% of lymphoma and 8.3% of myeloma, 66.9% for relapse, 4% for detectable MRD, 18.2% for mixed chimerism and 10.7% for high relapse risk. The overall response rate (CR and PR) was 28% (63% for mixed chimerism cases, 100% for detectable MRD cases). The median OS was 10.4 months [10]. More recently, Schneidawind reported a study of 44 patients mostly treated for AML and MDS, with curative, pre-emptive or prophylactic indication; patients treated with gDLI had a better outcome compared to patients treated with classical DLI in terms of chimerism conversion (75% vs. 25%, *p* = 0.006), 1 year cumulative incidence of relapse (46% vs. 70%, *p* = 0.04) and 1-year OS (46% vs. 70%, *p* = 0.04), without excess of GVHD [27]. The differences between our results and the latter publication may be explained by treatment associated with DLI or outcomes in the prophylactic situation.

## 5. Conclusions

Our results suggest similar effectiveness and toxicity of gDLI and classical DLI when treating or preventing clinical or molecular relapse or mixed chimerism. Those results from retrospective data should be interpreted with caution, and a randomized prospective trial including haploidentical allo-SCT is required.

## Figures and Tables

**Figure 1 jcm-09-02204-f001:**
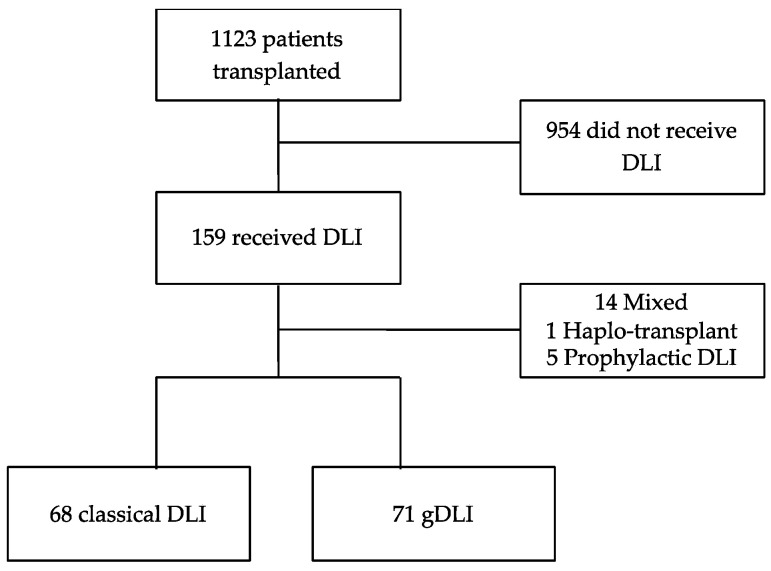
Flow chart.

**Figure 2 jcm-09-02204-f002:**
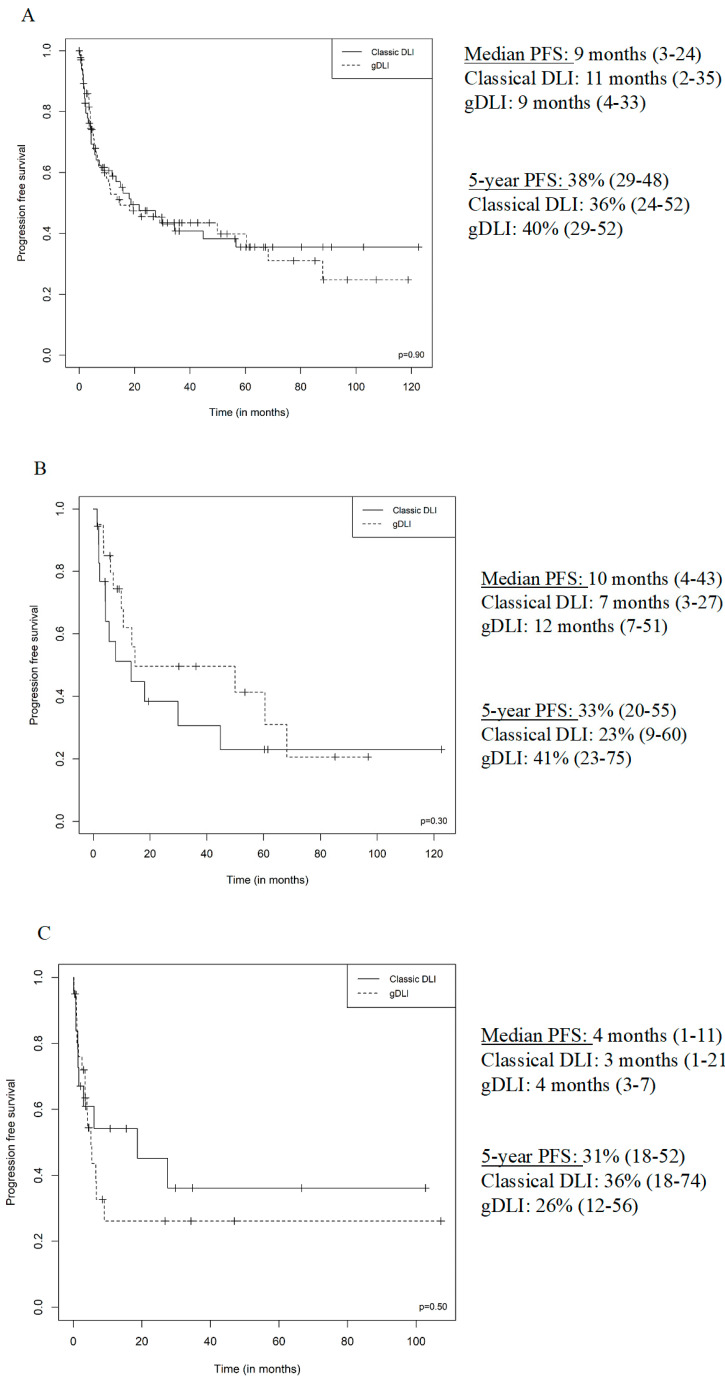
PFS from first DLI according to DLI type (Classical, gDLI), (**A**) all patients, (**B**) multiple myeloma, (**C**) acute myeloid leukaemia, (**D**) relapse indication, (**E**) pre-emptive indication.

**Figure 3 jcm-09-02204-f003:**
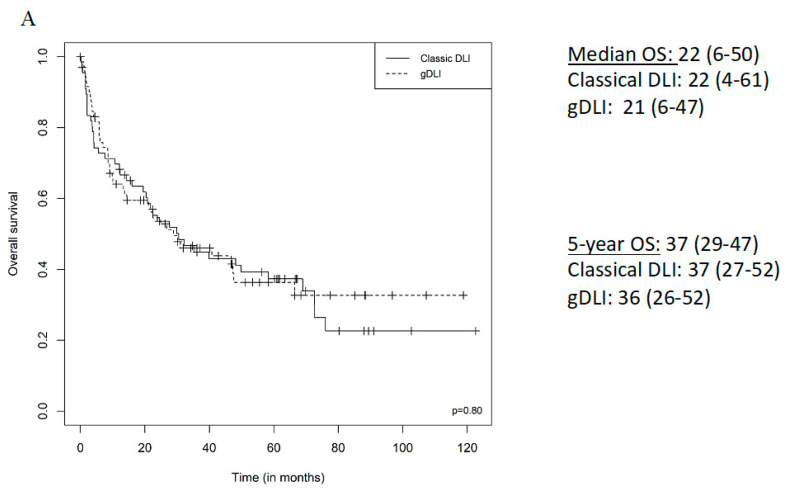
OS from first DLI according to DLI type (Classical, gDLI), (**A**) all patients, (**B**) multiple myeloma, (**C**) acute myeloid leukaemia, (**D**) relapse indication, (**E**) pre-emptive indication.

**Table 1 jcm-09-02204-t001:** Patients characteristics, *n* = 139.

	All DLI *n* = 139	Classical DLI *n* = 68 (48.9%)	gDLI *n* = 71 (51.1%)	*p*
Age at DLI, median (IQR)	52 (41–61)	49 (24–58)	55 (47–63)	<0.01
Men, *n* (%)	76 (54.7)	39 (57.4)	37 (52.1)	0.61
Disease, *n* (%)				0.05
Acute lymphoblastic leukaemia	20 (14.4)	15 (22.1)	5 (7.1)	
Acute myeloblastic leukaemia	45 (32.4)	20 (29.4)	25 (35.2)	
Hodgkin lymphoma	8 (5.8)	3 (4.4)	5 (7.1)	
Chronic lymphocytic leukaemia	7 (5.0)	6 (8.8)	1 (1.4)	
Chronic myeloid leukaemia	2 (1.4)	1 (1.5)	1 (1.4)	
Non-Hodgkin Lymphoma	10 (7.2)	3 (4.4)	7 (9.9)	
Myeloma	38 (27.3)	18 (26.5)	20 (28.2)	
Osteomyelofibrosis	1 (0.7)	0	1 (1.4)	
Myelodysplasia	8 (5.8)	2 (2.9)	6 (8.5)	
Status at transplant, *n* (%)				0.95
Complete remission	76 (54.7)	37 (54.4)	39 (54.9)	
Partial remission	38 (27.3)	18 (26.5)	20 (28.2)	
Stable or progressive disease	25 (18.0)	13 (19.1)	12 (16.9)	
Conditioning, *n* (%), *n* = 138			*n* = 70	0.02
Myeloablative conditioning	32 (23.2)	22 (32.4)	10 (14.3)	
Reduced intensity conditioning	92 (66.7)	42 (61.8)	50 (71.4)	
Sequential	14 (10.1)	4 (5.9)	10 (14.3)	
Donor type, *n* (%)				<0.001
Related	68 (48.9)	44 (64.7)	24 (33.8)	
Unrelated	71 (51.1)	24 (35.3)	47 (66.2)	
HLA compatibility, *n* (%)				0.30
10/10	122 (87.8)	62 (91.2)	60 (84.5)	
9/10	17 (12.2)	6 (8.8)	11 (15.5)	
Cell source, *n* (%)				<0.0001
Bone marrow	24 (17.3)	24 (35.3)	0	
Peripheral hematopoietic stem cells	115 (82.7)	44 (64.7)	71 (100)	
CD34 cells (×10^6^), median, (IQR)	5.0 (4.4–5.5)	4.7 (4.0–5.5)	*n* = 70 5.0 (5.0–5.4)	0.02
CD3 cells (×10^7^), median (IQR)	16.9 (9.6–23.0)	16.9 (5.5–29.8)	*n* = 70 16.9 (10.8–20.6)	0.99
GvHD, *n* (%)				0.86
All grades of acute GvHD	111(79.9)	48 (70.6)	63 (88.7)	
Grade 1 acute GvHD	80 (57.6)	37 (54.4)	43 (60.6)	
Grade 2 acute GvHD	21 (15.1)	5 (7.4)	16 (22.5)	
Grade 3 acute GvHD	9 (6.5)	5 (7.4)	4 (5.6)	
Grade 4 acute GvHD	0	0	0	
Chronic GvHD, *n* (%)	22 (15.8)	15 (22.1)	7 (9.9)	0.04
3-month chimerism, *n* (%), *n* = 138		*n* = 67		0.39
100% donor	56 (40.6)	30 (44.8)	26 (36.6)	
Mixed	81 (58.7)	37 (55.2)	44 (62.0)	
Recipient	1 (0.7)		1 (1.4)	
Indication for DLI 1, *n* (%)				0.84
Pre-emptive: mixed chimerism	22 (15.8)	11 (16.2)	11 (15.5)	
Pre-emptive: detectable minimal residual disease	21 (15.1)	9 (13.2)	12 (16.9)	
Relapse	96 (69.1)	48 (70.6)	48 (67.6)	
Number of DLIs, *n* (%)				0.24
<3	105 (75.5)	48 (70.6)	57 (80.3)	
≥3	34 (24.5)	20 (29.4)	14 (19.7)	
Period, *n* (%)				<0.0001
1998–2004	18 (12.9)	18 (26.5)	0	
2005–2011	59 (42.4)	33 (48.5)	26 (36.6)	
2012–2018	62 (44.6)	17 (25.0)	45 (63.4)	
Associated treatment with DLI, *n* = 133		*n* = 62		0.14
Yes	46(34.6)	17 (27.4)	29 (40.8)	
No	87(65.4)	45 (72.6)	42 (59.2)	

Abbreviations: gDLI = g-csf primed DLI, DLI = donor lymphocyte infusion, IQR = interquartile range, HLA = human leukocyte antigen, GvHD = graft versus host disease.

**Table 2 jcm-09-02204-t002:** DLI characteristics, *n* = 268.

	DLI 1	DLI 2	DLI 3	DLI 4	DLI 5	DLI 6	DLI 7
Patients, *n* (%)	139 (100)	67 (48.2)	34 (24.5)	15 (10.8)	7 (5.0)	5 (3.6)	1 (0.7)
Term, median from transplant (IQR) (month)	8.5 (5.7–15.5)	14.4 (11.3–24.1)	27.1 (18.0–65.1)	48.6 (26.6–71.5)	60.2 (58.4–68.0)	71.6 (71.2–72.9)	73.3 (NA)
CD3^+^, median (IQR) (10^7^/kg)	1 (0.5–1)	1.4 (1–5)	2.2 (1–6.6)	3.4 (1.8–5.5)	5 (1.1–10)	2.5 (1.2–5.2)	1 (NA)

Abbreviations: DLI = donor lymphocyte infusion, IQR = interquartile range.

**Table 3 jcm-09-02204-t003:** Patient outcomes, *n* = 139.

	All DLI *n* = 139	Classical DLI *n* = 68 (48.9%)	gDLI *n* = 71 (51.1%)	*p*
Complete response, *n* (%), *n* = 131		*n* = 64	*n* = 67	0.28
Yes	50 (38.2)	21 (32.8)	29 (43.3)	
No	81 (61.8)	43 (67.2)	38 (56.7)	
Post-DLI GvHD, *n* (%), *n* = 137		*n* = 66		
All grades of GvHD	64 (46.7)	30 (45.5)	34 (47.9)	0.86
Treatment-requiring GvHD, *n* (%)	32 (23.4)	14 (21.2)	18 (25.4)	0.81
Chimerism, *n* (%), *n* = 127		*n* = 61	*n* = 66	0.63
100% donor	88 (69.3)	43 (70.5)	45 (68.2)	
Mixed	38 (29.9)	17 (27.9)	21 (31.8)	
Recipient	1 (0.8)	1 (1.6)		
Progression-free survival				
Median PFS (months, IQR)	9 (3–35)	11 (2–35)	9 (4–33)	0.90
5-year PFS (%, IQR)	38 (29–48)	36 (24–52)	40 (29–56)	
Overall survival,				
Median OS (months, IQR)	22 (6–50)	22 (4–33)	21 (6–47)	0.50
5 years of OS (%, IQR)	37 (29–47)	37 (27–52)	36 (26–52)	
Cause of Mortality, *n* (%), *n* = 82			*n* = 70	0.14
Progression	65 (47.1)	37 (54.4)	28 (40)	
Toxicity	14 (10.1)	5 (7.4)	9 (12.9)	
Other	3 (2.2)	0	3 (4.3)	

Abbreviations: gDLI = g-csf primed DLI, DLI = donor lymphocyte infusion, GvHD = graft versus host disease, IQR = interquartile range, OS = overall survival, PFS = progression-free survival.

**Table 4 jcm-09-02204-t004:** Univariate and multivariate logistic regression response analysis to DLI treatment, *n* = 131.

	Univariate OR (95% CI)	*p*	Multivariate OR (95% CI) *	*p*
Age at DLI1 >60 years, *n* = 38	0.94 (0.42–2.08)	0.88		
Gender				
Female	1 (ref)	-		
Male (*n* = 76)	0.82 (0.41–1.68)	0.59		
Disease			14.07 (4.05–59.57)	<0.0001
Hodgkin Lymphoma	2.12 (0.41–10.99)	0.37		
Acute lymphoblastic leukaemia	1.01 (0.29–3.45)	0.99		
Acute myeloblastic leukaemia	1 (ref)	-		
Chronic lymphocytic leukaemia	2.12 (0.41–10.99)	0.37		
Chronic myeloid leukaemia	2.83 (0.16–49.22)	0.48		
Non-Hodgkin lymphoma	11.28 (2.07–61.44)	<0.01		
Multiple myeloma	1.67 (0.63–4.40)	0.30		
Osteomyelofibrosis	2.71 × 10^−4^ (3.59 × 10^−91^–2.04 × 10^83^)	0.94		
Myelodysplasia	8.46 (1.48–48.26)	0.02		
Indication for DLI				
Pre-emptive	1 (ref)	-		
Relapse	0.21 (0.09–0.45)	<0.0001	0.18 (0.07–0.45)	<0.001
Associated treatment				
No	1 (ref)			
Yes	0.59 (0.27–1.27)	0.18		
Type of DLI				
Classical	1 (ref)	-		
gDLI	1.56 (0.77–3.18)	0.22		ns
Post DLI GvHD			8.53 (3.01–26.78)	<0.0001
No GvHD	1 (ref)			
GvHD not requiring treatment	1.91 (0.78–4.69)	0.16		
Treatment-requiring GvHD	9.08 (3.28–25.14)	<0.0001		

Abbreviations: DLI = donor lymphocyte infusion, gDLI = g-csf stimulated donor lymphocyte infusion, OR = odd ratio, CI = confidence interval, GvHD = graft versus host disease. * Multivariate model for indication comparing non-Hodgkin lymphoma and myelodysplasia vs. others, model for indication comparing relapse and pre-emptive situation vs. others and model for GvHD comparing treatment-requiring GvHD vs. others.

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
