# Peer review of "A Retrospective Comparison of DLI and gDLI for Post-Transplant Treatment"

_jcm, 2020, doi:10.3390/jcm9072204_

Round 1

Reviewer 1 Report

The manuscript is revised according to reviewer’s comment and acceptable for publication.

Author Response

Dear reviewer,

Thanks you for your appreciation. We are responding to another reviewer questions.

  • We mentioned the reason for exclusion of haplo-HSCT from the analyze in the discussion section (page 2, line 67).
  • We added that we found no autologous recovery in the results (page 3, line 132).
  • The question of mixed chimerism patients is very interesting. We made a lot of subgroup analysis that wern't planned initially to explore hypothesis. The sub-group with mixed chimerism indication includes 22 patients, 11 had gDLI, 11 had classical DLI. Their outcome is quite good: 78% had chimerism conversion, median OS was 33 months. There were no differences between groups (p=0.47). We choose not to publish those results because they are not relevant due to reasons mentioned above (small sample, and non-planned analyze), and because we already have a lot of supplementary data and sub-groups analyses.

I hope those explanations will meet your expectation.

Kinds regards,

Sylvain Lamure

Reviewer 2 Report

Sylvain Lamure and colleagues reported on a large retrospective study on the use of DLi and Gdli in post HSCt patients.

This paper is well written and of interest to the Haematology community.

I believe it deserves to be published.

Some minor recommendations:

  • add an explanatory sentence on the exclusion of haploidentical HSCT.
  • an explanatory table on chimerism and outcome should be created.
  • When discussing chimerism the possibility of autologous recovery should at least be mentioned, even if no cases where identified.

Author Response

Dear reviewer,

Thanks you for your appreciation. To respond to your suggestions:

  • We mentioned the reason for exclusion of haplo-HSCT from the analyze in the discussion section (page 2, line 67).
  • We added that we found no autologous recovery in the results (page 3, line 132).
  • The question of mixed chimerism patients is very interesting. We made a lot of subgroup analysis that wern't planned initially to explore hypothesis. The sub-group with mixed chimerism indication includes 22 patients, 11 had gDLI, 11 had classical DLI. Their outcome is quite good: 78% had chimerism conversion, median OS was 33 months. There were no differences between groups (p=0.47). We choose not to publish those results because they are not relevant due to reasons mentioned above (small sample, and non-planned analyze), and because we already have a lot of supplementary data and sub-groups analyses.

I hope those explanations will meet your expectation.

Kinds regards,

Sylvain Lamure

This manuscript is a resubmission of an earlier submission. The following is a list of the peer review reports and author responses from that submission.

Round 1

Reviewer 1 Report

In this manuscript, Lamure S, et al. reported a retrospective comparative study of conventional DLI and G-CSF stimulated DLI (gDLI) in larger populations. This study demonstrated no significant differences in efficacy and safety between the DLI and gDLI groups. Although the manuscript is not acceptable for publication in Journal of Clinical Medicine in its present form, I would like to reconsider the manuscript if major revisions are made.

Major points

In this study, patients receiving haploidentical allogeneic HSCT were excluded. However, gDLI is a potential effective therapeutic option for haploidentical allogeneic HSCT. In the Abstract and Discussion section, you should not conclude no significant differences in efficacy and safety between the DLI and gDLI groups, without any comments for haploidentical allogeneic HSCT.

In the Introduction section, you described that there are multiple biological effects of G-CSF on peripheral T cells. However, you did not demonstrate any data to compare the difference in cell content or biological function between DLI and gDLI.

In the multivariate analysis for response rate, severe post-DLI GvHD was significantly associated with higher response rate, however, approximately 10% of all patients died due to toxicity of the DLI (GvHD and infection). You should perform multivariate analyses (including the type of DLI, and severity of post-DLI GVHD) to PFS, OS, or TRM. 

Minor points

Methods, patients and procedure (page 2, line 64): how did you examine donor chimerism status?

Results, patient and DLI characteristics (page 3, line 108): what did “14” mean?

In table 2, Residual disease (not residual disease)

Did you examine the difference in the incidence or severity of cytopenia after DLI and gDLI, which is one of the most severe DLI-related complications.

Author Response

Dear Reviewer 1,

Thanks you for taking time to review this manuscript. You can find our modification detailled in the following text.

Major points

In this study, patients receiving haploidentical allogeneic HSCT were excluded. However, gDLI is a potential effective therapeutic option for haploidentical allogeneic HSCT. In the Abstract and Discussion section, you should not conclude no significant differences in efficacy and safety between the DLI and gDLI groups, without any comments for haploidentical allogeneic HSCT.

Thanks for underlying this point. Due to small number of patients who underwent Haplo all-SCT and received DLI (N=1), we excluded those from the analysis. We specified this point in the abstract. We added a comment in the discussion, restricting the interpretation to allo-SCT from HLA matched related donor or unrelated HLA matched or mismatched (9/10) donor; and in the perspective we mention that this question should be studied for Haplo trasnplant.

In the Introduction section, you described that there are multiple biological effects of G-CSF on peripheral T cells. However, you did not demonstrate any data to compare the difference in cell content or biological function between DLI and gDLI.

This is a very interesting point, however, despite many attempts, we haven’t been able to carry out a biological project. I hope that we can study those points in the future, in a prospective multicentric trial. A sentence had been appended in the discussion section, considering that point (page 14, line 3).

In the multivariate analysis for response rate, severe post-DLI GvHD was significantly associated with higher response rate, however, approximately 10% of all patients died due to toxicity of the DLI (GvHD and infection). You should perform multivariate analyses (including the type of DLI, and severity of post-DLI GVHD) to PFS, OS, or TRM. 

Thanks for the suggestion. We performed the analyses. Data shows that toxicity related mortality (TRM), combining GvHD and infections post-DLI occurred in 10.1% of the patients. Univariate and multivariate analysis shows association of occurrence of treatment requiring GvHD and TRM, HR =9.6 (2.1-43.7, p<0.01) (Supplementary table 1). Univariate analysis shows association of ALL, CLL, NHL and MM with a better OS, relapse indication for DLI was associated with a worse OS, HR=5.06 (2.7-9.6, p<0.01) and post DLI GvHD not requiring treatment was associated with a better OS, HR=0.5 (0.29-0.87, p=0.01). This last point was not confirmed in the multivariate analysis (supplementary table 2). Univariate analysis shows association of NHL a better PFS, HR=0.23 (0.07-0.76, p=0.02), relapse indication for DLI was associated with a worse PFS, HR=3.31 (1.87-5.86, p<0.01) and post DLI GvHD not requiring treatment and treatment-requiring GvHD were associated with a better PFS, HR=0.56 (0.33-0.94, p=0.03) and HR=0.19 (0.08-0.45, p<0.01). All those associations remained significant in the multivariate analysis (supplementary table 3). Those data were included in the manuscript (TRM page 9 line 17 : TRM, OS page 10, line 3 and PFS page 9 line 16).

Minor points

Methods, patients and procedure (page 2, line 64): how did you examine donor chimerism status?

Chimerism was determined using the multiplex amplification of short tandem repeat markers and fluorescence detection. We added a mention in the method section and a reference (page 4, line 22, ref 8).

Results, patient and DLI characteristics (page 3, line 108): what did “14” mean?

This was a mistake, from an older version of the manuscript that included 14 patients that have been treated successively with gDLI and classical DLI.

In table 2, Residual disease (not residual disease)

I don’t find this point.

Did you examine the difference in the incidence or severity of cytopenia after DLI and gDLI, which is one of the most severe DLI-related complications?

Cytopenia has been described as one of the major toxicities following DLI. However, this wasn’t a concern in our experience. We observed some mild and transient cytopenia. We decided not to report those. A sentence had been appended in the discussion section, considering that point (page 13, line 25).

We hope those revisions will clarify the manuscript, allowing it for publication.

Kind regards,

S LAMURE

Reviewer 2 Report

The authors compare outcomes after gDLI and classical DLI after stem cell transplant for residual disease and mixed chimerism. The article is well thought out, and well written. There are no concerns with the English of the manuscript and no concerns with methods. Additionally, the conclusions are supported by the results of the study. I have no suggestions or concerns with the manuscript as written.

Author Response

Dear reviewer 2,

Thanks you for taking time to review this manuscript, and thanks you for your nice comment.

Best regards,

S LAMURE

Reviewer 3 Report

In this retrospective study, Lamure et al. compared the efficacy and safety of gDLI versus classical DLI in a cohort with hematologic malignancies. 

1)  Based on the method (line 60) this retrospective study was done on patients who underwent HLA matched allo-HCT. However, based on the results (line 116) near 12% of patients received mismatched transplant. Please clarify whether you included MMUD or not. Since authors excluded Haplo transplant, it is unclear why MMUD was included.   

2) Please clarify whether you are talking about acute or chronic GVHD in line 157 and what is the definition of severe GVHD

3) It is unclear if any of the patients were on any immunosuppression at the time of DLI. Please indicate the percentage of the patients who were on any immunosuppression at the time of DLI. 

4) Suggest clarifying what percentage of  classical DLI was fresh versus cryopreserved. If there is adequate numbers, recommend comparing the results of fresh versus cryopreserved classical DLI since cryopreservation has shown to alter graft content.

5) Suggest clarifying the definition of response for patients who received DLI for "mixed chimerism"? Achieving complete donor chimerism? What is the definition of response (line 161)

6) Based on the results (line 169), primary disease (non-Hodgkin lymphoma) predict response to DLI. The numbers are too small to make any conclusion (only 3 patients with non-Hodgkin received classical DLI). It may be more useful to used DRI (disease risk index) rather than individual disease to evaluate response to DLI.

7) Table 3. Please clarify cause of death  "toxicity" and  "others". Rather than Toxicity, it is more useful to know more granular data such as  the percentage of patients who died of GVHD or any other specific toxicity.  

8) Limitations  inherent to a retrospective study need to be mentioned in the discussion. 

Author Response

Dear Reviewer 3,

Thanks you for taking time to review this manuscript. You can find our responses in the following text.

1)  Based on the method (line 60) this retrospective study was done on patients who underwent HLA matched allo-HCT. However, based on the results (line 116) near 12% of patients received mismatched transplant. Please clarify whether you included MMUD or not. Since authors excluded Haplo transplant, it is unclear why MMUD was included.  

Thanks for this point. We did included MMUD. I did correct the mistake in the method section. Due to small number of patients who received DLI after haplo-identical allo-SCT in our center during the study (N=1), we decided to exclude those from the study. This has been specified in the methods section (page 4, line 16).

2) Please clarify whether you are talking about acute or chronic GVHD in line 157 and what is the definition of severe GVHD

As characterization of GvHD postDLI from medical charts is difficult, we chosed to mash up acute GvHD and overlap syndrome of chronic GvHD, as mentioned in the methods section. We specified when GvHD required treatment. (page 5 line 6).

3) It is unclear if any of the patients were on any immunosuppression at the time of DLI. Please indicate the percentage of the patients who were on any immunosuppression at the time of DLI. 

In our center all patient that had DLI indication should have had immunosuppressive treatment interruption a few weeks before the injection.  We specified this point in the method section (page 5 line 4).

4) Suggest clarifying what percentage of classical DLI was fresh versus cryopreserved. If there is adequate numbers, recommend comparing the results of fresh versus cryopreserved classical DLI since cryopreservation has shown to alter graft content.

All patients who had classical DLI had fresh collected cells for their 1st injection. Some patients in this group received DLI2 from cryoconserved aliquots of the excess of the 1st apheresis, the others had DLI2 with fresh cells from another apheresis etc. Taken together there are many possible combinations, and analysis would be difficult. This point could be assessed in a prospective study.

5) Suggest clarifying the definition of response for patients who received DLI for "mixed chimerism"? Achieving complete donor chimerism? What is the definition of response (line 161)

Yes, response aimed for patients with mixed chimerism is full donor chimerism conversion. This point is specified in the methods section (page 5, line 10).

6) Based on the results (line 169), primary disease (non-Hodgkin lymphoma) predict response to DLI. The numbers are too small to make any conclusion (only 3 patients with non-Hodgkin received classical DLI). It may be more useful to used DRI (disease risk index) rather than individual disease to evaluate response to DLI.

Disease risk index miss from a majority of medical charts, due to the age of some data, and the initial treatment is performed in different hospital from our region. Those missing data makes analysis impossible.

7) Table 3. Please clarify cause of death "toxicity" and "others". Rather than Toxicity, it is more useful to know more granular data such as the percentage of patients who died of GVHD or any other specific toxicity.  

Toxicity related mortality is combining death due to GvHD or infection following DLI, as those situations frequently overlaps. We specified this point in the methods section (page 5 line 17) and added a “Safety” section in the results (page 9 line 36).

8) Limitations inherent to a retrospective study need to be mentioned in the discussion. 

We appended a sentence in the conclusion “Those results from retrospective data should be interpreted with caution, and randomized prospective trial, including haploidentical allo-SCT is required.”

I hope those revisions are clarifying the manuscript, allowing it to be published.

Kind regards,

S LAMURE

Round 2

Reviewer 1 Report

In this manuscript, Lamure S, et al. reported a retrospective comparative study of conventional DLI and G-CSF stimulated DLI (gDLI) in larger populations. Although the manuscript is not acceptable for publication in Journal of Clinical Medicine in its present form, I would like to reconsider the manuscript if major revisions are made.

Major points

In the Discussion section, the arguments are not laid out clearly; this section needs rewriting for clarity.

This article has a lot of grammatical errors, especially, lack of conjuctions and commas in many sentences. Therefore, I will not consider that the manuscript is acceptable for this journal, until the author(s) should ask someone familiar with the English language to help you rewrite the paper.